# The Impact of Tocilizumab on Radiological Changes Assessed by Quantitative Chest CT in Severe COVID-19 Patients

**DOI:** 10.3390/jcm11051247

**Published:** 2022-02-25

**Authors:** Ana-Maria-Jennifer Anghel, Cristian-Mihail Niculae, Eliza-Daniela Manea, Mihai Lazar, Mara Popescu, Anca-Cristina Damalan, Adela-Abigaela Bel, Iulia-Maria Nedelcu, Raluca-Elena Patrascu, Adriana Hristea

**Affiliations:** 1National Institute for Infectious Diseases Prof. Dr. Matei Bals, No. 1, Calistrat Grozovici Street, Sector 2, 021105 Bucharest, Romania; amya.anghel@gmail.com (A.-M.-J.A.); eliza.manea@drd.umfcd.ro (E.-D.M.); mihai.lazar@umfcd.ro (M.L.); scoicaru.anca@gmail.com (A.-C.D.); bel.adela@yahoo.com (A.-A.B.); iulia-maria.nedelcu@drd.umfcd.ro (I.-M.N.); raluca.jipa1@drd.umfcd.ro (R.-E.P.); adriana.hristea@umfcd.ro (A.H.); 2Infectious Diseases Department, Faculty of Medicine, University of Medicine and Pharmacy Carol Davila, No. 37, Dionisie Lupu Street, Sector 2, 020021 Bucharest, Romania; 3Faculty of Life Sciences and Medicine, King’s College London, 5th Floor Addison House, Guy’s Campus, London WC2R 2LS, UK; popescu.mara@gmail.com

**Keywords:** quantitative chest CT, severe COVID-19, tocilizumab timing

## Abstract

(1) Background: We aimed to analyze the characteristics associated with the in-hospital mortality, describe the early CT changes expressed quantitatively after tocilizumab (TOC), and assess TOC timing according to the oxygen demands. (2) Methods: We retrospectively studied 101 adult patients with severe COVID-19, who received TOC and dexamethasone. The lung involvement was assessed quantitatively using native CT examination before and 7–10 days after TOC administration. (3) Results: The in-hospital mortality was 17.8%. Logistic regression analysis found that interstitial lesions above 50% were associated with death (*p* = 0.01). The other variables assessed were age (*p* = 0.1), the presence of comorbidities (*p* = 0.9), the oxygen flow rate at TOC administration (*p* = 0.2), FiO_2_ (*p* = 0.4), lymphocyte count (*p* = 0.3), and D-dimers level (*p* = 0.2). Survivors had a statistically significant improvement at 7–10 days after TOC of interstitial (39.5 vs. 31.6%, *p* < 0.001), mixt (4.3 vs. 2.3%, *p* = 0.001) and consolidating (1.7 vs. 1.1%, *p* = 0.001) lesions. When TOC was administered at a FiO_2_ ≤ 57.5% (oxygen flow rate ≤ 13 L/min), the associated mortality was significantly lower (4.3% vs. 29.1%, *p* < 0.05). (4) Conclusions: Quantitative imaging provides valuable information regarding the extent of lung damage which can be used to anticipate the in-hospital mortality. The timing of TOC administration is important and FiO_2_ could be used as a clinical predictor.

## 1. Introduction

In patients hospitalized with severe and critical COVID-19, corticosteroids (CS) and immunomodulatory agents are essential therapeutic resources in order to block the cytokine storm. Tocilizumab (TOC) is a humanized monoclonal antibody that targets the IL-6 receptor, thus stopping the inflammatory process. Data from randomized clinical trials (RCTs), including patients with severe COVID-19, overall support the use of CS, but are inconsistent for TOC. Some RCTs found that TOC does not change mortality, disease progression, or secondary infection rate (COVACTA, EMPACTA, BACC-Bay) [1,2,3], while others found an increase in survival rate, lower progression to mechanical ventilation (MV), and decrease in time to discharge, with no difference in secondary infection rates (CORIMUNO, RECOVERY, REMAP-CAP) [4,5,6]. A meta-analysis that included 27 RCTs found a significant mortality benefit in co-administering TOC and CS in patients with respiratory support, but not in those under MV, with better outcomes when given early in the course of the disease. It also showed that TOC and CS decreased progression to MV, but did not significantly impact the secondary infection rate [7]. While data suggest TOC was beneficial in the treatment of COVID-19 patients, it is unclear on what the optimum timing should be. In addition, these important studies involving patients treated with TOC did not include in the analysis a quantification of the radiological impact of the anti-IL-6 therapy [1,2,3,4,5,6]. Other observational studies evaluating patients treated with TOC recorded changes in chest X-ray (CXR) [8,9,10]. There is limited data on CT scan changes and improvement at 7 or 14 days after TOC administration and the existing data included a limited number of patients [11,12].

We aimed to analyze the radiologic changes expressed quantitatively associated with the risk of in-hospital death among severe COVID-19 patients treated with CS plus TOC, we also aimed to describe the CT changes at 7-10 days after TOC administration.

The secondary objective was to assess if the timing of TOC administration according to the oxygen flow rate impacts the in-hospital mortality among severe patients receiving CS plus TOC.

## 2. Materials and Methods

### 2.1. Study Design and Population

This was a retrospective, observational cohort study including consecutive severe adult patients admitted to our department between March 2020 and January 2021. The study was approved by the local ethics committee. We included hospitalized adult patients (≥18 years old), with confirmed COVID-19 (positive SARS-CoV-2 RT-PCR), severe disease (defined as having at least one of the following criteria: peripheral oxygen saturation (SpO_2)_ ≤ 93% in ambient air, respiratory rate (RR) > 30/min, arterial oxygenation partial pressure to fractional inspired oxygen ratio (PaO_2_/FiO_2_ ratio) < 300, or lung infiltrates > 50% of lung parenchyma) [13], treated with the standard of care (supplemental oxygen, antivirals, anticoagulants), antibiotics if indicated, and dexamethasone, plus at least one dose of TOC.

All patients who had a contraindication for TOC were excluded: active neoplasia, neutropenia (neutrophils less than 1000/mmc), severe thrombocytopenia (platelets less than 50,000/mmc), active Mycobacterium tuberculosis infection, viral hepatitis B and C [14]. We also excluded patients who received another immunomodulatory agent (IL-1 inhibitors), patients already included in other studies, pregnant and breastfeeding women.

TOC was given to adult patients who required supplemental oxygen (through nasal cannula, Venturi mask, non-rebreathing mask, high-flow oxygen therapy, or mechanical ventilation, both invasive and non-invasive), in order to maintain normal SpO_2_ values. TOC was given intravenously at 4–8 mg/kg body weight (up to a maximum of 800 mg/dose) in one to three slow infusions set 12–24 h apart. Patients were closely monitored during the infusion. TOC was given off-label, according to the National Guidelines for COVID-19 management from the Ministry of Health and the decision for its administration was at the discretion of the patient’s clinician and upon availability in the hospital. Patients agreed to the use of TOC and were included after signing the informed consent form. We used DXM in doses of 8 up to 24 mg (0.5–1 mg/kg) for a median of 11 days, including tapering. The antiviral therapies used were Remdesivir, when available, at the dose of 200 mg in day 1, then 100 mg/day for 5–7 days, and Umifenovir 200 mg three times daily for 5–7 days.

### 2.2. Outcomes

We collected demographic variables (age and sex), comorbidities information (arterial hypertension, diabetes mellitus, coronary artery disease, congestive heart failure, obesity, asthma, chronic obstructive pulmonary disorder, cancer, immunosuppression, chronic kidney disease) and patient’s duration of symptom prior to hospital presentation. Respiratory function was evaluated by measuring the RR, SpO_2_ by pulse oximetry and/or SpO_2_ by arterial blood gasses analysis, along with PaO_2_ and PaO_2_/FiO_2_ ratio. The laboratory data were collected at the moment of TOC administration and 72 h after and included: complete blood count with differential, coagulation data (D-dimer level), inflammation markers (C-reactive protein—CRP, ferritin, IL-6), and serum biochemical data (alanine transaminase and lactate dehydrogenase). Adverse events relating to TOC were also recorded. Standard 12-lead electrocardiography was performed for all patients at the moment of hospital admission.

The severity of lung involvement was assessed using native CT examination. Imaging evaluation was done at the moment of TOC administration and 7–10 days after. All the scans were performed on a 64 slice CT Somatom Definition As (Siemens); we used spiral acquisitions (with breath-hold when possible) with a pitch of 1.2, CARE Dose4D and CARE kV active in order to reduce the radiation dose, 1.2 mm collimation and 3 mm reconstructions in both mediastin (B31f image filter) and lung window (B80f ultra sharp image filter). For the quantitative evaluation we used Syngo Pulmo 3D, which allowed us to segment the lung parenchyma (excluding the main pulmonary vessels, trachea and main bronchi from the densitometric assessment) and to calculate the lung involvement percent based on specific densitometric intervals. In our study we considered alveolar lesion for densities higher than 0 Hounsfield units (HU), mixt lesions (alveolar and interstitial) for densities between 0 and −200 HU, interstitial lesions for densities between −200 and −800 HU, normal parenchyma for densities between −800 and −1000 HU, and hyperinflation or emphysema for densities lower than −1000 HU. The imaging review was blinded and the radiologist was unaware of the treatment patients received.

### 2.3. Statistical Analysis

We analyzed the collected data using the Statistical Package for Social Sciences (SPSS version 21, IBM Corp., Armonk, NY, USA). Data were expressed as medians and IQR for continuous variables and as frequencies and percentages for categorical variables. Continuous variables were compared using the Mann—Whitney U test. The chi-square test or Fisher’s exact test was used to compare categorical variables. A *p*-value of <0.05 was considered statistically significant. Multivariable logistic regression analysis was performed to analyze the relationship between mortality and age, presence of comorbidities, oxygen flow requirement at TOC administration, lymphocyte count, D-dimer levels, and interstitial lung changes above 50%.

## 3. Results

### 3.1. Clinical, Laboratory Data and Radiologic Changes at TOC Administration

During the study period, 567 patients were hospitalized in one department of our institution, with 187 having severe COVID-19. Of those, 101 consecutive patients received TOC and were followed up in this study. All those excluded were not administered TOC either because of contraindications or TOC supply gaps.

In our cohort, 79 (78.2%) patients were male and the median age was 61 (51–67) years. Comorbidities were seen in 79 (78.2%) patients and consisted of obesity (55.6%), type 2 diabetes (25.3%), hypertension (59.4%), heart failure (7.5%), peripheral vascular disease (2.5%), chronic kidney disease (3.79%), chronic obstructive pulmonary disease (13.9%), chronic hepatitis (3.7%), history of neoplasia (5%), ischemic stroke (5%), dementia (2.5%), peptic ulcer (1.2%), and other pathologies (46.8%) including history or current depressive disorder and rheumatologic diseases. The median Charlson Comorbidity Index (CCI) score was 2 (1–3).

A total of 18 (17.8%) patients were receiving supplemental oxygen by nasal cannula or Venturi mask, 81 (80.19%) patients by non-rebreathing masks, and one patient was mechanically ventilated at the moment of receiving TOC. The oxygen flow had a median of 14 (10–22) L/min, the FiO_2_ median was 60 (50–60)%, the PaO_2_/FiO_2_ ratio had a median of 144 (117–194), and the respiratory rate had a median of 26 (21–30) breaths/min.

Before TOC administration, the main changes in the laboratory parameters recorded as median (IQR) were: lymphocyte count 800 (600–1200) cells/mm^3^, neutrophils/lymphocytes ratio 8.1 (4.5–14.2), CRP 81.5 (33.4–147.5) mg/L, LDH 367 (303–460) U/L, D-dimers 243 (155–472) ng/mL, and ALT 45 (30.5–76.5) U/L. Ferritin and IL-6 measurements were performed in 60% of the patients and the medians (IQR) were 1046.35 (354.2–1514.1) ng/mL and 9.25 (1.82–99.32) pg/mL, respectively.

All patients had thorax CT scan changes suggestive of consolidation, mixt lesions, and interstitial lesions, assessed by volumetric measurement based on HU ranges and expressed as a percentage from the total lung volume. All patients had interstitial lesions in at least two lobes and 96% of them had lesions in all five lobes. Consolidating lesions were present in 36% of patients and accounted mostly for lesions in two lobes in 15.8% of the CT scans. Atelectatic changes were observed in 86.1% of patients, with changes predominantly in two, three, four and five lobes in 22.7%, 13.8%, 18.8% and 26.3% patients, respectively. The median number of lobes affected by atelectatic lesions was 3 (2–3).

In-hospital mortality in our study was 17.8%. According to the in-hospital mortality, two groups were created: a first group (survivors) including 83 patients and a second group (non-survivors) with 18 patients. Clinical and laboratory data divided by outcome are shown in Table 1.

In total, 19% of the patients received Remdesivir, for a median duration of 5 days, of which 13 (15%) patients were in the survivor’s group and five (27%) patients were in the non-survivors group (*p* = 0.18). A total of twenty-nine patients received Umifenovir, 25 in the survivor’s group and four in the non-survivor’s group (*p* = 0.5).

In all patients TOC was given at a median of nine (7–11) days since symptoms onset. The total dose of TOC was 1620 (1200–2400) mg in the survivor’s group versus 1440 (1200–1800) mg in patients who died. Patients were given TOC at a median of 9 (7–11) days since symptoms onset versus 8 (4–9) days, respectively.

A total of nine patients received TOC after intensive care unit (ICU) admission, of which eight patients died. Twelve patients were admitted in the ICU after TOC administration: three patients for COVID-19 related thrombosis and eight patients for worsening in respiratory function 24–72 h after TOC. Twenty patients had a 1–5-fold increase in liver enzymes, two had hypofibrinogenemia, one reported paresthesia and facial flushing, and one had an episode of hypotension, no serious adverse events were encountered.

COVID-19 related complications noticed in this cohort were: pulmonary embolisms (five patients), acute coronary syndrome (one patient), pulmonary edema (one patient), acute lower extremity ischemia (one patient), and suspicion of myocarditis (two patients).

The rate of secondary infection was 9.9%, including three *Clostridioides difficile* infections. In 5 (4.45%) patients infections occurred after ICU admission. 

Between the two groups, there was a statistically significant difference regarding age, presence of comorbidities, CCI, lymphocyte count, D-dimers, the median percentage of lung parenchyma affected by interstitial lesions, mixt lesions, and alveolar consolidation.

The logistic regression found that the median percentage of interstitial lesions expressed quantitatively above 50% was associated with death (*p* = 0.01). The other variables assessed were age (*p* = 0.1), the presence of comorbidities (*p* = 0.9), the oxygen flow rate at TOC administration (*p* = 0.2), lymphocyte count (*p* = 0.3), and D-dimers level (*p* = 0.2).

### 3.2. Comparative Radiologic Changes before and after TOC Administration 

The comparative analysis of radiologic changes between the survivors and non-survivors at the moment of TOC administration and 7–10 days after are summarized in Table 2.

In patients who survived, there was a statistically significant improvement of interstitial, mixt and consolidating lesions between the moment of TOC administration and 7–10 days after. In the patients who died, there was a progression of interstitial lesions without statistical significance.

The decrease of mixt and consolidating lesions after TOC administration in non-survivors shows the positive impact of TOC at limiting the progression of the inflammatory process, however it is insufficient to control the high intensity of the inflammatory process.

### 3.3. Timing of TOC Administration According to the Oxygen Demands at the Time of Administration

To assess if the timing of TOC impacts the in-hospital mortality based on the oxygen demands at the time of administration, we used the ROC curve to determine the cut-off value of oxygen flow, FiO_2_, and PaO_2_/FiO_2_. The area under the curve for oxygen flow, FiO_2_, and PaO_2_/FiO_2_ was 0.708 (CI: 0.564–0.851), 0.608 (CI: 0.460–0.756), and 0.508 (CI: 0.350–0.666), respectively. The cut-off value for the oxygen flow rate was 13 L/min, and the cut-off value for FiO_2_ was 57.5%, corresponding to an oxygen flow rate of 12–13 L/min. An oxygen flow rate greater than 13 L/min (or FiO_2_ greater than 57.5%) was associated with a significantly higher mortality (4.7% vs. 27.6%, *p* = 0.03). We divided the studied population in two groups: 46 patients with FiO_2_ ≤ 57.5% (oxygen flow rate ≤ 13 L/min) and 55 patients with FiO_2_ > 57.5% (oxygen flow rate > 13 L/min). The comparative analysis between the two groups is shown in Table 3.

There was no significant difference between the two groups regarding age, gender, number and nature of the comorbidities.

The duration of symptoms until TOC administration varied from 0 to 24 days (median of 9 days). Approximately 80% of patients had a duration from symptoms onset to TOC administration for less than 10 days in both groups.

Patients requiring oxygen flow >13 L/min (FiO_2_ > 57.5%) had a lower lymphocyte count, PaO_2_/FiO_2_ ratio, higher CRP, and more severe interstitial, mixt and consolidating CT changes.

Although without statistical significance, the respiratory rate, neutrophils/lymphocyte ratios, D-dimers, LDH, and ALT had higher values in patients requiring oxygen flow >13 L/min (FiO_2_ > 57.5%).

## 4. Discussion

In this study we analyzed clinical laboratory parameters and radiological changes in 101 severe COVID-19 patients who received TOC therapy in addition to the standard CS therapy at different points during their treatment.

The RECOVERY trial, a large RCT, has shown TOC to be an effective treatment in hospitalized COVID-19 patients, reducing the 28-day mortality and leading to quicker discharge rates [5]. Similar to the in-hospital mortality in our study (17.8%), a meta-analysis of 34 studies in which TOC was used reported a mortality of 16% [15]. Another meta-analysis of 10 observational studies involving 1358 patients found a lower mortality for those treated with TOC compared to patients who did not receive anti-IL-6 therapy (17.1% vs. 27.6%) [16]. In another study involving 1351 patients, 20% of patients in the standard care group died compared to only 7% in the TOC group, with neither group receiving CS [17]. Importantly, in another study, a significant mortality benefit was only found when anti-IL-6 therapy was co-administered with CS [7].

Consistent with findings from other studies, patients with severe forms and unfavorable outcomes were male, older (median age of 66 versus 58 years old) and had more comorbidities [18,19]. They also required significantly higher oxygen flow rates when the first dose of TOC was given.

At the time of TOC administration all patients had a hyperinflammatory state characterized by increased levels of IL-6, CRP, ferritin, LDH and/or D-dimers, as well as lymphopenia [8,12]. While some inflammatory markers, such as the D-dimers and the lymphocyte count demonstrated significant differences between survivors and non-survivors, there was no statistically significant difference in the CRP and LDH values. Previous studies have established the neutrophil to lymphocyte ratio as a marker of disease severity, with neutrophil count increasing as lymphocyte count decreases [20]. In our cohort the non-survivor’s group demonstrated higher values compared to the survivors, although the differences were not statistically significant. The average neutrophils/leukocytes ratio was 8.1 (4.5–14.2) which tends to be found in those with critical illness [21] and is much higher than the cut-off of over 6.5 used to predict mortality [8].

Shen et al. described a CXR scoring system. It has been shown that the initial CXR scores have prognostic value and are associated with ICU admission, and MV, but not with the length of hospitalization, duration of MV, or mortality [22]. While it did not quantify the extent of lung involvement, a previous small study also found differences between the CT score of the survival and mortality group, with significantly more patients in the mortality group having consolidations and air bronchogram [23].

We assessed the degree of lung injury using a CT scan and we found that the interstitial changes of more than 50% were significantly associated with the risk of death. Alveolar condensation and mixt lesions, although affecting less than 5% and 10% of the lung, respectively, were more extended in non-survivor patients. The interstitial changes were significantly (*p* < 0.001) higher in the non-survivor’s group; 52 (46.7–60.3) versus 39.5 (31.6–48.4). There is limited data regarding the optimal timing for the administration of anti-IL-6 therapy based on radiological findings and imaging changes after TOC administration. In our study, at 7–10 days post TOC administration, those patients with favorable outcomes had shown significant radiological improvement, while in the unfavorable outcome group there were no significant changes (Table 2). The regression of interstitial, mixt and consolidating lesions was observed in 57 (70%), 29 (35%), and 24 (29%) patients, respectively.

Similar to the findings we noted on CT scans, Toniati et al. [8] and Moreno et al. [9] reported improvement in lung lesions as assessed by CXR. At 10 days after administration of TOC 61% and 67.9% of cases had improved, respectively.

In a retrospective cohort study, 30 patients receiving TOC for COVID-19 pneumonia were matched with 27 other patients treated only with antivirals/antimalarials. CT chest scans on admission and at 14 days were compared using a combined semiquantitative and texture analysis of all parenchymal lesions. While there was no significant difference in the chest CT scan score in the TOC group (mean value of 14.5 ± 4.8 vs. 12.3 ± 4, *p* = 0.09), some texture features significantly changed, including mean attenuation, skewness, and entropy, suggesting a considerable fading of pulmonary parenchymal lesions. Patients not receiving TOC showed a significant increase in both chest CT score (mean value of 8.8 ± 4.4 vs. 12.5 ± 4.1, *p* = 0.01) and several texture features [11]. In one small prospective study involving 42 patients the authors found a significant improvement in pulmonary CT scan imaging 14 days after TOC, in 66% of patients [12]. 

Regarding the optimal timing of TOC administration affecting mortality, there is some evidence that TOC administration can be beneficial in ICU patients, particularly if started within 24 h of admission, potentially by reversing organ damage before it becomes permanent [6]. Although in our study eight out of nine (88.9%) patients receiving TOC after ICU admission died, admissions were influenced by bed availability which could have negatively impacted our results. In a cohort of 112 patients hospitalized with severe COVID-19, comparing patients who received TOC within 10 days (median of 8 days, IQR 7–10) or after 10 days (median of 13 days, IQR 12–16) from symptoms onset, the authors found a statistically significant difference in the 90-day mortality (18.6% vs. 5%), while the 30-day mortality was not significantly different between the two groups [9].

In a retrospective study, the use of symptoms onset as a marker of disease severity might have limitations due to biased reporting. As such, we proposed the use of FiO_2_/oxygen flow rates as an objective measure that could assist decisions on when TOC should be administered. Using the ROC curve, we calculated the cut-off point of oxygen flow rates of 13 L/min (FiO_2_ of 57.5%) as the markers to decide on TOC administration. The in-hospital mortality of patients who received TOC when the oxygen flow rate was ≤13 L/min (FiO_2_ ≤ 57.5%) was significantly lower vs. patients who received TOC when they required > 13 L/min (FiO_2_ > 57.5%), 2 (4.7%) vs. 16 (27.6%). Most patients received TOC within 10 days from symptoms onset. 

A prospective meta-analysis also found IL-6 antagonist administration associated with lower death rates and progression to MV. At 28 days, fewer patients receiving an oxygen flow rate < 15 L/min at randomization had progressed to invasive MV, extracorporeal membrane oxygenation or died as compared to those receiving noninvasive ventilation [24].

In our study, the group receiving less oxygen when TOC was given showed less radiological lung involvement and considerably lower CRP values. While it can be argued that patients requiring less oxygen have less severe forms of the disease and are likely to have better outcomes independent of treatment, early administration of TOC could be contributing to the improved outcomes.

The strengths of our study include the proposed use of oxygen flow rate requirements or FiO_2_ as a cut-off for administering TOC which provides an objective measure of disease severity. While there is limited data on radiological changes after the administration of TOC, we used chest CT scanning to quantify the degree of lung involvement and to dynamically assess improvement. Higher CT scores were correlated with increased hospital mortality and, at 7–10 days after TOC administration, the survivors had significant radiological improvement.

As it was a retrospective, observational, non-randomized study it has several limitations, with it being prone to confounding factors. There was an imbalance in the proportion of genders, with 77.2% of patients being male and the median age was 61 years. As such, it remains unclear what the ideal timing of TOC administration should be for other demographics. Additionally, we were unable to account for the different strains of SARS-CoV-2 or the genetic makeup of our patients during our analysis. A randomized controlled trial will be required to further investigate the timing and effects of TOC. Other limitations of our study include the lack of data on 30 and 90-day mortality as the patient follow-up was only in hospital.

## Figures and Tables

**Table 1 jcm-11-01247-t001:** Patient characteristics and laboratory findings, according to the outcome, recorded at TOC administration.

	Survivors N = 83	Non-Survivors N = 18	*p* Value OR
Clinical and Laboratory Data			
Male, N (%)	64 (77.1)	14 (77.8)	0.9
Age (years), median (IQR)	58 (50–65)	66 (62–71)	0.03
Presence of comorbidities, N (%)	61 (73.5)	18 (100)	0.01 OR 0.7 (0.6–0.8)
Charlson Comorbidity Index, median (IQR)	2 (1–3)	3 (2–5)	<0.001
Respiratory rate (breaths/minute), median (IQR)	26 (21–30)	28 (21–31)	0.08
Oxygen flow rate (L/min), median (IQR)	13 (10–16)	15 (14–28)	0.008
FiO_2_ (%), median (IQR)	60 (50–60)	60 (60–60)	0.4
PaO_2_/FiO_2_ ratio, median (IQR)	145 (118–190)	140 (105–207)	0.7
$\mathrm{Lymphocyte}\;\mathrm{count}\;(\mathrm{cells}/{\mathrm{mm}}^{3}$), median (IQR)	900 (600–1300)	600 (500–1000)	0.02
Neutrophils/lymphocytes ratio, median (IQR)	7.5 (4.5–13.1)	14.2 (7.2–18.3)	0.1
CRP (mg/L), median (IQR)	78 (31–140)	91 (40–173)	0.7
LDH (U/L), median (IQR)	362 (296–430)	447 (323–563)	0.1
D-dimers (ng/mL), median (IQR)	235 (153–386)	367 (172–735)	0.009
ALT (U/L), median (IQR)	45 (30–77)	52 (30–76)	0.9

IQR = interquartile range, CRP = C reactive protein, LDH = lactat dehydrogenase, ALT = alanine aminotransferase.

**Table 2 jcm-11-01247-t002:** Radiologic changes at 7–10 days after TOC administration, according to the outcome.

Radiologic Changes	Survivors N = 83		*p* Value	Non-Survivors N = 18		*p* Value
	Before TOC	After TOC		Before TOC	After TOC	
Interstitial lesions (%), median (IQR)	39.5 (31.5–48.4)	31.6 (24.7–41.2)	<0.001	52 (46.7–60.3)	57.7 (42.7–62.7)	0.9
Mixt lesions (%), median (IQR)	4.3 (2.5–7.2)	2.3 (1.5–3.8)	0.001	8.2 (4.1–13.7)	6 (3–8.7)	0.7
Consolidating lesions (%), median (IQR)	1.7 (1–3.1)	1.1 (0.6–1.6)	0.001	3.1 (1.5–7)	2.4 (1.6–4.8)	0.2

IQR = interquartile range.

**Table 3 jcm-11-01247-t003:** Comparative analysis according to the oxygen flow rate and FiO_2_ at TOC administration.

	FiO_2_ ≤ 57.5% (Oxygen Flow Rate ≤ 13 L/min) N = 46	FiO_2_ > 57.5% (Oxygen Flow Rate > 13 L/min) N = 55	*p* Value
Age (years), median (IQR)	57 (50–64)	61 (52–68)	0.2
Male sex, N (%)	37 (80.4)	42 (76.3)	0.6
Comorbidities, N (%)	36 (78.2)	43 (78.1)	0.9
Charlson Comorbidities Index, median (IQR)	2 (1–2)	2 (1–4)	0.09
Symptoms ≤ 11 days, N (%)	36 (78.2)	46 (79.3)	0.6
Duration of symptoms until TOC administration (days), median (IQR)	9 (7–11)	9 (6.5–11)	0.1
Respiratory rate (breaths/minute), median (IQR)	24 (20–30)	27 (22–33.5)	0.1
PaO_2_/FiO_2_ ratio median (IQR)	155 (130–217)	133 (102–173.5)	0.02
Lymphocyte count (cells/mm^3^), median (IQR)	1053 (700–1500)	808 (537–1000)	0.01
Neutrophils/lymphocyte ratio, median (IQR)	5.4 (4–11.3)	10 (5.1–15.9)	0.2
CRP (mg/L), median (IQR)	45.1 (19.9–86.3)	95.8 (45.5–178)	<0.001
D-dimers (ng/mL), median (IQR)	229 (129–367)	261 (156–637)	0.1
LDH (U/L), median (IQR)	344 (264.7–409.7)	390 (325–489)	0.09
ALT (U/L), median (IQR)	58 (31.5–98.5)	71 (29.2–73.5)	0.2
Interstitial lesions (%), median (IQR)	36.1 (27.3–45.1)	47.3 (37.6–55.5)	<0.001
Mixt lesions (%), median (IQR)	3.7 (1.6–4.7)	7.65 (3.8–10.6)	<0.001
Consolidation (%), median (IQR)	1.4 (0.8–2.1)	2.9 (1.2–5.7)	<0.001
Deaths, N (%)	2 (4.3)	16 (29.1)	0.03

IQR = interquartile range, TOC = tocilizumab, CRP = C reactive protein, LDH = lactat dehydrogenase, ALT = alanin aminotransferase.

## Data Availability

The data presented in this study are available on request from the corresponding author.

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
