# Peer review of "The Impact of Tocilizumab on Radiological Changes Assessed by Quantitative Chest CT in Severe COVID-19 Patients"

_jcm, 2022, doi:10.3390/jcm11051247_

Round 1

Reviewer 1 Report

This study has a design flaw. The impact of Tocilizumab on radiological changes can not be adequately analyzed without a control group not receiving the drug.

The proposed timing of Tocilizumab administration based on oxygen flow rate is interesting but undoubtedly requires a more detailed statistical analysis to convince the readers.

Reviewer 2 Report

①2. Materials and Methods 2.1. Study design and population

If there are mutations or differences in viral strains during the course of the study, should differences in response to TOC and timing of administration depending on the strain be considered?

②3. Results 

86 cases are excluded from 187 cases, which is too many. What is the reason?

When comparing different oxygen delivery methods, isn't it more appropriate to compare FiO2 rather than oxygen flow rate?

The median oxygen flow rate is shown to be 14 L/min, but FiO2 may vary depending on the method of oxygen delivery (nasal canulae, HNF, MV).

The oxygen flow rate of 13 L/min measured by ROC as a cutoff value for TOC administration should be presented by FiO2.

3.2. Comparative radiologic changes before and after TOC administration

In Table 2, about the change in CT findings in fatal and non-fatal patients before and after TOC administration, isn't it the same as the difference in imaging findings between patients who survive after severe illness and those who do not, regardless of TOC?

3.3. Timing of TOC administration according to the oxygen flow rates

As mentioned above, the cutoff value is O2 13L/min, but the oxygen volume is difficult to interpret because FiO2 varies depending on the method of oxygen delivery.

Round 2

Reviewer 1 Report

Thank you for improving this paper, especially regarding the timing of TOC administration.

Still, I disagree with the title of the paper, "The impact of tocilizumab on radiological changes...". Without the control group, this paper does not provide evidence that the radiological changes are indeed the result of TOC.

Therefore I would suggest the authors to change the title of their paper. Something in line with "Optimal timing of TOC administration" would seem more appropriate.

Reviewer 2 Report

The authors adequately addressed my concerns.